# Differences in Bacterial Small RNAs in Stool Samples from Hypercholesterolemic and Normocholesterolemic Subjects

**DOI:** 10.3390/ijms24087213

**Published:** 2023-04-13

**Authors:** Cristian Morales, Raul Arias-Carrasco, Vinicius Maracaja-Coutinho, Pamela Seron, Fernando Lanas, Luis A. Salazar, Nicolás Saavedra

**Affiliations:** 1Centro de Biología Molecular y Farmacogenética, Núcleo Científico-Tecnológico en Biorecursos BIOREN, Universidad de La Frontera, Temuco 4811230, Chile; 2Tecnología Médica, Facultad de Salud, Universidad Santo Tomás, Temuco 4801076, Chile; 3Programa Institucional de Fomento a la Investigación, Desarrollo e Innovación, Universidad Tecnológica Metropolitana, Santiago 8330378, Chile; 4Advanced Center for Chronic Diseases—ACCDiS, Facultad de Química y Ciencias Farmacéuticas, Universidad de Chile, Santiago 8380494, Chile; 5Departamento de Ciencias de La Rehabilitación, Facultad de Medicina, Universidad de La Frontera, Temuco 4781151, Chile; 6Departamento de Medicina Interna, Facultad de Medicina, Universidad de La Frontera, Temuco 4781151, Chile; 7Departamento de Ciencias Básicas, Facultad de Medicina, Universidad de La Frontera, Temuco 4811230, Chile

**Keywords:** bacterial small RNAs, gut microbiota, cholesterol metabolism, small RNA sequencing

## Abstract

Cholesterol metabolism is important at the physiological level as well as in several diseases, with small RNA being an element to consider in terms of its epigenetic control. Thus, the aim of this study was to identify differences between bacterial small RNAs present at the gut level in hypercholesterolemic and normocholesterolemic individuals. Twenty stool samples were collected from hypercholesterolemic and normocholesterolemic subjects. RNA extraction and small RNA sequencing were performed, followed by bioinformatics analyses with BrumiR, Bowtie 2, BLASTn, DESeq2, and IntaRNA, after the filtering of the reads with fastp. In addition, the prediction of secondary structures was obtained with RNAfold WebServer. Most of the small RNAs were of bacterial origin and presented a greater number of readings in normocholesterolemic participants. The upregulation of small RNA ID 2909606 associated with *Coprococcus eutactus* (family *Lachnospiraceae*) was presented in hypercholesterolemic subjects. In addition, a positive correlation was established between small RNA ID 2149569 from the species *Blautia wexlerae* and hypercholesterolemic subjects. Other bacterial and archaeal small RNAs that interacted with the LDL receptor (LDLR) were identified. For these sequences, the prediction of secondary structures was also obtained. There were significant differences in bacterial small RNAs associated with cholesterol metabolism in hypercholesterolemic and normocholesterolemic participants.

## 1. Introduction

Cholesterol is an essential biomolecule for cells and human physiology [1]. Among its functions, the maintenance of the stability of the cell membrane stands out, and so do acting as a precursor for different hormones and producing bile, among others [2,3]. Among the different types of cholesterol, the one most associated with cardiovascular diseases (CVDs), particularly with atherosclerosis, is hypercholesterolemia for low-density lipoprotein (LDL) cholesterol (C), so correct control is essential to avoid this group of diseases [4,5]. The regulation of cholesterol levels is carried out with various mechanisms, with the regulation of biosynthesis being one of the main ones, and two key elements to consider are sterol regulatory element-binding protein 2 (SREBP2) and 3-hydroxy-3-methyl-glutaryl coenzyme A reductase (HMGCR) [1]. To these are added transcription factors such as those associated with the liver X receptor (LXR), retinoid X receptors (RXRs), and a receptor of great importance, the LDL receptor (LDLR), which are under the regulation of some small RNAs, such as miRNAs, which make it possible to control the gene expression of these molecules associated with cholesterol metabolism [6].

Small RNAs have been the subject of research in recent years and have been identified in different animals and plants, as well as in many other species and types of cells. Despite this, many of the functions, interactions, and impacts that most of these biomolecules may have on cells are still unknown. Their function is the epigenetic regulation of cellular processes, mainly through the silencing or repression of gene expression [7]. They belong to the group of non-coding RNAs (ncRNAs), which also includes long non-coding RNAs (lncRNAs), of more than 200 nucleotides (nt) in size. The main, or classic, small RNAs range from 18 to 31 nt in size, although they could be considered up to 150 nt, given the discoveries of new groups of small RNAs [7,8,9]. Typically, they are classified into microRNAs (miRNAs), PIWI-RNAs (piRNAs), small interference RNAs (siRNAs), transfer RNAs (tRNAs), small nucleolar RNAs (snoRNA), small nuclear RNAs (snRNAs), and circular RNAs (circRNAs). In the same way, new groups have appeared, such as small RNAs derived from RNA Y (ytRNAs), small RNAs derived from RNA vault (vtRNAs), and small RNAs derived from tRNA (tsRNAs) [8,9,10]. Regarding the latter, their presence in eukaryotic cells, archaea, and bacteria has already been corroborated [11].

In general, small RNAs participate in cellular homeostasis, regulating vital processes associated with gene expression through messenger RNAs (mRNAs) [12]. Their impact can be appreciated at different levels, e.g., in fundamental processes such as embryonic development and the innate immune response [13], as well as at the physiological level in humans, where some miRNAs and piRNAs have been associated with the development of gonads and spermatogenesis [14], among others. The presence of small RNAs can be commonly detected in various bodily fluids, such as blood (and its derivatives), saliva, or urine, in which they occur in different proportions [15]. In addition, in humans, small RNAs have been associated with various pathologies, such as some types of cancer [10], celiac disease [16], acute myocardial infarction (AMI) [17], and CVDs in general [18], among others. For this reason, some small RNAs that can be used as biomarkers and for interference (therapeutic level) to be applied in CVDs have already been proposed; however, more studies are needed to ensure their viability and feasibility of use [17,18,19].

Although each species presents small RNAs of its own, the interaction between the small RNAs of one species with a different species has also been evidenced, allowing microorganisms, plants, and animals to have a horizontal transfer of information [20]. This was experimentally demonstrated by Koeppen et al. (2016) using bacterial vesicles, derived from *Pseudomonas aeruginosa*, that contained a short RNA that managed to reduce the release of cytokines and the infiltration of neutrophils in cultured human cells and mice, respectively [21]. Similarly, it has also been shown that some small RNAs from *Helicobacter pylori* can reduce the release of interleukin 8 (IL-8) in cell cultures [22]. In a work prepared by Zhu et al. (2022), the presence of functional DNA in microbial vesicles from the microbiota of human feces was evidenced, which, according to functional analysis, presents a slight association with CVDs [23]. Although the above is about DNA, it demonstrates the transfer and interaction of the genetic material of microbiota with human host cells.

Considering the above, it has been proposed to use these interactions between microbiota and humans as biomarkers for common and priority diseases, such as cancer and obesity [24]. In addition, according to the above-described research, the use of RNA synthetic biology has been proposed for the creation and use of small RNAs for various therapeutic purposes, also considering the mechanism of delivery to target cells, to avoid the activation of the immune system or degradation by nucleases present in the blood, among others [25,26].

Considering the role of small RNAs in the organism and their interaction with various pathologies, this study aimed to identify differences between bacterial small RNAs present at the gut level among hypercholesterolemic and normocholesterolemic individuals.

## 2. Results

### 2.1. Anthropometric, Clinical and Biochemical Parameters

Anthropometric, clinical and biochemical parameters are summarized in Table 1. The mean age of the groups was not significantly different (*p* = 0.525). Similarly, there were no significant differences in SBP, DBP, glucose levels and HDL-C levels. Triglycerides (*p* = 0.003), total cholesterol (*p* < 0.0001) and LDL-C (*p* < 0.0001) were significantly higher in hypercholesterolemic individuals, while weight (*p* < 0.0001) and BMI (*p* = 0.004) were significantly higher in normocholesterolemic individuals.

The results presented above were originally described by Morales et al. (2022). From the hypercholesterolemic (*n* = 27) and normocholesterolemic (*n* = 30) groups, the 10 participants with the highest levels of LDL-C and the 10 participants with the lowest levels of this biomolecule were selected. The selection of the participants with the highest and lowest LDL-C levels was carried out to be able to associate the presence of small RNAs with the most extreme levels of LDL-C in the studied cohort, in order to clearly establish the differences in the expression of small RNAs in individuals with these conditions. It is not ruled out to characterize in the future the small RNAs present in the participants who have intermediate levels of LDL cholesterol to complement the results obtained in this work.

The selected hypercholesterolemic participants (*n* = 10) presented an average of 5.32 mmol/L (±0.79), while the normocholesterolemic participants (*n* = 10) presented 2.00 mmol/L (±0.58) of LDL-C.

### 2.2. Bioinformatic Analysis of Small RNAs Sequences

Once the sequencing process of the 20 selected participants (*n* = 10, hypercholesterolemic; *n* = 10, normocholesterolemic) was completed, bioinformatic analysis was performed. The average raw reads were 12.883.704, while the reads obtained after quality filtering processing were 11,189,004. The sum of all raw reads (*n* = 20) was 257,674,098. In the case of High-quality reads (*n* = 20), the sum was 223,780,087. The 86.85% of the total reads presented a Q30 quality score (99.9% base accuracy). When looking at each study group separately, there is a greater dispersion in terms of the number of sequences in hypercholesterolemic participants compared to normocholesterolemic participants. In the latter, a greater number of sequences can be seen on average, both unprocessed in Fastp (Raw) and already processed (High-Quality) (Figure 1).

The results presented above were originally described by Morales et al. (2022). From the hypercholesterolemic (*n* = 27) and normocholesterolemic (*n* = 30) groups, the 10 participants with the highest levels of LDL-C and the 10 participants with the lowest levels of this biomolecule were selected. The selection of the participants with the highest and lowest LDL-C levels was carried out to be able to associate the presence of small RNAs with the most extreme levels of LDL-C in the studied cohort, in order to clearly establish the differences in the expression of small RNAs in individuals with these conditions. It is not ruled out that the small RNAs present in the participants who had intermediate levels of LDL cholesterol could be characterized in the future to complement the results obtained in this work.

The selected hypercholesterolemic participants (*n* = 10) presented an average of 5.32 mmol/L (±0.79) LCL-C, while the normocholesterolemic participants (*n* = 10) presented 2.00 mmol/L (±0.58) LDL-C.

### 2.3. Bioinformatic Analysis of Small RNA Sequences

Once the sequencing process of the 20 selected participants (*n* = 10, hypercholesterolemic; *n* = 10, normocholesterolemic) was completed, bioinformatic analysis was performed. The average raw reads were 12,883,704, while the reads obtained after quality filtering processing were 11,189,004. The sum of all raw reads (*n* = 20) was 257,674,098. In the case of high-quality reads (*n* = 20), the sum was 223,780,087. In total, 86.85% of the total reads presented a Q30 quality score (99.9% base accuracy). When looking at each study group separately, there was a greater dispersion in terms of the number of sequences in hypercholesterolemic participants than in normocholesterolemic participants. In the latter, a greater number of sequences could be seen on average, both unprocessed, in Fastp (raw), and already processed (high quality) (Figure 1).

As a result of principal component analysis (PCA), it was observed that there were differences in the signature of small RNAs in the studied groups. Nine of the ten normocholesterolemic participants were clearly grouped, while the hypercholesterolemic participants presented a greater dispersion. Five of the ten hypercholesterolemic participants were clearly grouped and presented similarity, while four hypercholesterolemic participants were in the normocholesterolemic group. Regarding their LDL-C levels, these were 5.23 mmol/L, 5.46 mmol/L, 5.67 mmol/L, and 5.80 mmol/L, while the participant at the top of the figure had an LDL-C value of 5.34 mmol/L (Figure 2).

Among the 20 participants, the total read count assigned as small RNAs was 36,582,653 (16.34% of high-quality reads), with an average of 1,829,133 reads per sample. The number of small RNAs was equivalent to 142,319 in total (7115 on average per sample). When classifying small RNAs, it was obtained as a result that they were associated with bacteria (99,737), archaea (1912), and eukaryotes (humans) (5483), and unclassified (35,177).

In hypercholesterolemic participants, the small RNAs associated with bacteria were 47,560 (4756 on average), the largest number. Those that were unclassified corresponded to 16,508 (1650 on average); those from eukaryotes were 2603 (260 on average); and those from archaea were 905 (90 on average).

Regarding normocholesterolemic participants, the small RNAs associated with bacteria were 52,177 (5217 on average) and were the most prevalent, followed by those that were unclassified (18,669; 1866 on average), those from eukaryotes (2890; 289 on average), and finally, those from archaea (1007; 100 on average). More details are in Appendix A.

Figure 3 shows the average of the small RNAs of bacteria, archaea, and eukaryotes, and the unclassified ones, both for participants with hypercholesterolemia and for those with normocholesterolemia. There were no significant differences in the number of small RNAs classified in both studied groups.

Figure 4 shows the results of the differential expression analysis with an MA plot, where M (Y-axis) corresponds to the fold change (Log2 scale) of the small RNAs, and A (X-axis) corresponds to the expression level (Log10 scale) of the small RNAs from bacteria, archaea, and eukaryotes, and the unclassified ones. Most of the small RNAs were upregulated, although some had a high fold change value, except for one bacterial small RNA with an extremely downregulated fold change. A few small RNAs, mainly from bacteria, were highly expressed, with values equal to or greater than three (right side of the graph).

We then obtained the top 10 small RNAs with the lowest binding energy associated with each of the selected mRNA genes related to cholesterol metabolism, and most of them presented between one and two mRNAs as targets, equivalent to 69% of them (Figure 5). However, it was observed that one small RNA presented interactions with 18 of the 20 genes associated with cholesterol metabolism considered in this work, and interestingly, it was upregulated in hypercholesterolemic participants. This small RNA, with ID 2909606, is associated with the species *Coprococcus eutactus* (family *Lachnospiraceae*, class Clostridia), with 89% identity. By performing a recheck analysis with the entire BLAST nucleotide (nt) database, a new hit with 100% identity with the family *Lachnospiraceae* was obtained. More details are presented in Appendix A.

In addition, when performing the correlation between the LDL-C levels and the bacterial small RNAs obtained in our study using the Pearson coefficient, a positive correlation was obtained between the hypercholesterolemic participants and small RNA ID 2149569 associated with the bacterial species *Blautia wexlerae* DSM 19850 (*p* = 0.0013). This corresponds to the most significant result in terms of positive associations; more details can be found in Appendix A.

When analyzing the binding energy of small RNA 2909606, we found strong complementarity with the mRNA sequences of LIPC, NPC1, LDLR, SREBF1, ABCG1, NPC1L1, RXRA, ABCA1, PCSK9, ABCG8, APOB, CYP7A1, SREBF2, NR1H3, VLDLR, HMGCR, ABCG5, and APOC1. The strongest complementarity occurred with LIPC (−27.26), and the lowest complementarity, with APOC1 (−17.08). Figure 6 shows the binding energy levels associated with the complementarity of this small RNA with the rest of the mRNAs of genes associated with cholesterol metabolism. After LIPC, there was great complementarity with the mRNAs of NPC1 (−26.59), LDLR (−25.16), and SREBF1 (−25.02). The sequence of small RNA 2909606 presented a weaker interaction with APOA1 and APOE mRNAs than that found with respect to other mRNAs. So, they were not among the top of interactions.

Considering the relevance of the LDL receptor (LDLR) to people’s cholesterol levels, the small RNAs that interacted with the mRNA of this molecule were analyzed. The small RNAs with the best complementarity were obtained; the one that presented the best result among the nine was small RNA 2909606 (−25.16; *Coprococcus eutactus*/*Lachnospiraceae* family), followed by small RNA 676475 (−25.13), associated with *Phocaeicola vulgatus*. Other small RNAs were associated with the species *Bacterioides cellulosilyticus*, *Candidatus methanomethylophilus alvus*, *Geobacter* sp. SVR, *Streptomyces* sp. WY228, *Aneurinibacillus thermoaerophilus*, *Clostridium* sp. MD294, and *Bifidobacterium thermophilum*. All the indicated microorganisms are bacteria, except for *Candidatus methanemethylophilus alvus*, which belongs to the group of archaea. Figure 7 shows the IDs and binding energy levels associated with the complementarity of all the small RNAs that interacted with the mRNA LDLR.

### 2.4. Prediction of the Secondary Structure of Small RNAs That Bind to LDLR

From the MFE, the prediction of the secondary structure of the sequences of small RNAs that presented interaction with the mRNA LDLR was performed. In Figure 8, the nine molecules are presented, the representation of only one of which could not be obtained (Figure 8G), since the calculation of the representation obtained the MFE of 0.00 kcal/mol. The other small RNA sequences did have a predicted secondary structure, where some sequences had stem-loops at one end, while the other end was open (Figure 8A–D,H). In these cases, the MFE values were the following: 8A, −0.90 kcal/mol; 8B, −3.10 kcal/mol; 8C, −4.80 kcal/mol; 8D, −2.40 kcal/mol; 8H, −2.70 kcal/mol. In addition, sequences with stem-loops at both ends were found (Figure 8E,F,I), and these had the following MFE results: 8E, −7.70 kcal/mol; 8F, −6.00 kcal/mol; 8I, −3.90 kcal/mol.

## 3. Discussion

Considering the role of small RNAs at different levels in the organism, this work aimed to describe the significant differences in these biomolecules associated with cholesterol metabolism. For this, a case-control study was carried out with 10 hypercholesterolemic participants (5.32 mmol/L LDL-C on average) and 10 normocholesterolemic participants (2.00 mmol/L LDL-C on average). Of the total high-quality readings obtained, which were 223,780,087 (*n* = 20), 16.34% corresponded to small RNAs. This is because the RNA extraction and purification method applied to the stool samples obtained all these molecules, so in future studies, we will analyze these non-small RNA readings.

Regarding the small RNAs obtained, a greater number of reads was observed in normocholesterolemic participants, both raw and high-quality reads. In a work previously carried out by our group, Rebolledo et al. (2017) reported significant differences in terms of the richness, and the Shannon–Weaver and Simpson indices of bacterial communities of gut microbiota between normocholesterolemic and hypercholesterolemic individuals, with these values being notably decreased in the latter group [27]. Our results show that there were also differences between people with extreme levels of LDL-C in the amount of small RNAs present in the gut.

In both study groups, bacterial small RNA reads predominated compared with archaea, eukaryotic, and unclassified reads. These were also the most highly expressed in general. The above is to be expected, since this group of bacteria abounds in the human gut microbiota, so their biomolecules exist in greater quantity than those of other microbial groups. In addition, many of these molecules play an important role in the physiology and pathophysiology of some diseases. Such is the case of indole substances, intermediary substances that are metabolized by the liver into trimethylaminoxide (TMAO) and short-chain fatty acids (SCFAs), as well as lipopolysaccharides, which induce immunomodulation [28,29]. It has been established that only Bacteroides and Firmicutes represent 90% of the total microorganisms in the total composition at the gut level, followed by the phyla Actinobacteria, Proteobacteria, Verrucomicrobia, Cyanobacteria, and Fusobateria [30,31]. In BLASTn, our results showed 100% identity with the *Lachnospiraceae* family (89% for *C. eutactus*), which belongs to the Firmicute group.

Focusing on *C. eutactus*, the most recent studies associate it with butyrate metabolism in the intestine, where in the presence of D-β-hydroxybutyrate (DBHB), an increase in its abundance in the intestine has been observed [32]. Regarding DBHB, it has been reported that it has cellular mediator functions, influencing sensations of appetite and satiety, and functions in inflammatory processes, including neoplastic processes [33]. On the contrary, the decrease in the abundance of *C. eutactus* has been associated with the presence of blood in the stools [34], when compared with exposure to proinflammatory diets in Crohn’s disease [35], Parkinson’s disease [36], and irritable bowel syndrome (IBS) [37]. In addition, a new CRISPR-associated nuclease (Cas), called CeCas12a and belonging to the Cas12a nuclease family, has been associated with this species and is considered to have better characteristics than Cas9. Interestingly, Chen et al. (2020) demonstrated CeCas12a activity in human cells, thus opening new possibilities for research and therapeutic applications [38].

Our work associated the regulation of small RNA ID 2909606 with *C. eutactus* in hypercholesterolemic participants, a new sequence associated not only with this species, but also with the prediction of interaction with the LDLR. In general, several small RNAs associated with the epigenetic control of the LDLR have been reported, and we highlight, for example, miRNA-140-5P (LDLR of human hepatocytes), miRNA-185 and miRNA-128-1 (which directly affect the LDLR), miRNA-27a and -b (associated with the liver), and miRNA-148a (rodent liver regulation) [6,39]. In addition, regarding hypercholesterolemic participants, we were able to determine a positive correlation (using the Pearson coefficient) between small RNA ID 2149569 from the bacterium *B. wexlerae* DSM 19850 with this study group (*p* = 0.0013). *B. wexlerae* is a bacterium reported to be common in human feces [40], although its decrease has been reported in children with obesity and insulin resistance [41]. Hosomi et al. (2022) has reported an inverse correlation between the presence of *B. wexlerae*, and obesity and type 2 diabetes, showing metabolic improvements and anti-inflammatory effects in murine models [42]. Other studies have revealed the association between this species and weight loss in individuals that present the above in abundance at the gut level, considering it as a good predictor of weight loss [43]. Interestingly, this study group presented lower weight and BMI than normocholesterolemic participants (*p* < 0.0001).

Our results show strong complementarity between small RNA 2909606 and other mRNAs of genes associated with cholesterol metabolism, such as LIPC, NPC1, SREBF1, ABCG1, NPC1L1, RXRA, ABCA1, PCSK9, ABCG8, APOB, CYP7A1, SREBF2, NR1H3, VLDLR, HMGCR, ABCG5, and APOC1. We observed the strongest complementarity with LIPC mRNA (−27.26) and the lowest complementarity with APOC1 (−17.08). Bhattarai et al. (2021), in a review work, established the main small RNAs associated with the epigenetic control of cholesterol metabolism genes, with some of the most important ones being miRNA-33a and miRNA-27a for ABCA1; miRNA-33a and miRNA-128-2 for ABCG1; miRNA-27a and miRNA-128-2 for RXR; miRNA-27a for APOA1, APOB100 and APOE; miRNA-1 and miRNA-206 for SREBF1; miRNA-185 for SREBF2; and miRNA-223 for HMGCS1 [6]. Synthetic small inference RNAs (siRNAs) against PCSK9 have also been created therapeutically, such as Inclisiran, which supports the use of statins in patients with elevated LDL-C levels [44,45].

Most of the small RNAs associated with the control of lipid metabolism are of endogenous origin; however, other small RNAs of exogenous origin have also been reported. In a study conducted by Tarallo et al. (2022), it was shown that the profiles of small RNAs are different among omnivorous, vegetarian, and vegan people, and miRNA-425-3p and miRNA-638 were increased when bacteria of the genera *Roseburia* sp. and *Akkermansia muciniphila* were present [46]. At the level of cholesterol metabolism, a recognized example is miRNA-168a, of plant origin, which has been shown to regulate the expression of LDLR adapter protein 1 (LDLRAP1) [47]. Our results provide more evidence of epigenetic control by exogenous small RNAs of bacterial type in cholesterol metabolism. In addition to small RNA 290960, we obtained as a result other small RNAs that interacted with the mRNA LDLR, with those being IDs 676475 (*Phocaeicola vulgatus*), 2473147 (*Bacterioides cellulosilyticus*), 1622429 (*Candidatus methanemethylophilus alvus*), 2119964 (*Geobacter* sp. SVR), 2634900 (*Streptomyces* sp. WY228), 1456166 (*Aneurinibacillus thermoaerophilus*), 233262 (*Clostridium* sp. MD294), and 3196507 (*Bifidobacterium thermophilum*). For these small RNAs to exert some kind of effect on human cells in which cholesterol metabolism takes place, it is necessary that they be transported to them. For this, various mechanisms have been postulated with which bacterial small RNAs can finally reach and interact with human cells, with extracellular vesicles (EVs) standing out as one of the most studied [48]. EVs provide safe transport of microbial nucleic acids to other kingdoms, currently known as “inter-kingdom communication” [49]. EVs allow small RNAs such as miRNAs to tolerate the low pH associated with digestion, as well as the destructive mechanisms of intracellular phagolysosomes [50], eventually being able to come into contact with cells and regulate their gene expression.

By predicting the secondary structure of these bacterial small RNAs, an initial characterization of the molecules can be obtained to establish their similarities and differences with respect to other small RNAs, both endogenous and exogenous to humans. Our results provide secondary structure predictions for eight of the nine analyzed sequences, as we did not obtain a structure for small RNA 233262.

## 4. Materials and Methods

### 4.1. Subjects and Samples

A case-control study was designed with 10 hypercholesterolemic individuals and 10 normocholesterolemic individuals from a total cohort of 57 people who corresponded to the participants described by Morales et al. (2022) [51].

For the analysis of the biochemical parameters, samples of total venous blood without anticoagulant were obtained. For this, the standard venipuncture technique was used. In addition, stool samples were requested of the participants, and the samples were issued and delivered as soon as possible to the laboratory to be frozen at −80 °C. To ensure better viability of the nucleic acids of these samples, DNA/RNA Shield-Fecal Collection Tube (Zymo Research, Irvine, CA, USA) was used.

### 4.2. Anthropometric, Clinical, and Biochemical Parameters

Anthropometric and clinical parameters such as age, height, weight, body mass index (BMI), systolic blood pressure (SBP), and diastolic blood pressure (DBP) were obtained. Biochemical measurements were performed on serum obtained from total venous blood. Glucose, triglycerides, total cholesterol, high-density lipoprotein (HDL) cholesterol (C), and LDL-C were quantified using enzymatic colorimetric methods. Quality measurements were controlled using normal and pathological commercial serums (Wiener Lab., Rosario, Argentina).

The obtained data were statistically analyzed using the software Prisma v. 8.0 (GraphPad Software Inc., San Diego, CA, USA), with which descriptive statistics were obtained for each variable of the data of all included participants. In addition, the biochemical and anthropometric characteristics of the groups were compared with the Mann–Whitney test.

For all statistical analyses, significance was established at *p*-values less than 0.05.

### 4.3. RNA Extraction and Sequencing

A total of 250 mg of stools from the participants was used for total RNA extraction. For this, a ZR Soil/Fecal RNA MicroPrep™ kit (Zymo Research) was used following the manufacturer’s instructions, which not only allows one to obtain long RNAs but also makes it possible to obtain small RNAs. The RNA obtained was quantified with fluorometry using Quantus Fluorometer™ (Promega, Madison, WI, USA) and analyzed with capillary electrophoresis using Fragment Analyzer™ (Agilent Technologies, Santa Clara, CA, USA). For the qualification of samples, RNA had to have an RIN equal to or greater than 7.0. Subsequently, libraries were prepared using a TruSeq Small RNA Library Prep™ kit (Illumina, San Diego, CA, USA) following the manufacturer’s instructions. The libraries were quantified with fluorometry using the Quantus Fluorometer™ (Promega, USA) and analyzed with capillary electrophoresis using Agilent 2100 Bioanalyzer™ (Agilent Technologies, USA). Finally, sequencing was performed with HiSeq 50SE (Illumina, USA) according to the manufacturer’s instructions.

### 4.4. Bioinformatic Analysis

Raw reads were filtered using Fastp v0.23.2 [52] using the Illumina adapters provided by the FastQC tool [53]. First, a sliding window was applied with a quality higher than 28 and a size of 4 nucleotides; if the mean quality was lower than 30, the read was discarded. Finally, all reads with lengths greater than 15 nucleotides were conserved as high-quality reads.

The identification of small RNA candidates was performed using BrumiR [54], which is an automatic tool for the detection of small RNAs in next-generation sequencing (NGS) datasets based on the Bruijn graph. The parameters were the standards recommended by the authors. The final small RNA repository was the aggregation of two output files, candidate_miRNA.fasta and other_sequences.txt, which had miRNAs and other small RNA candidates, respectively.

To identify the read counts of each small RNAs for each sample, a mapping procedure was performed using Bowtie 2 [55]. Initially, through Bowtie2-build, the indices were generated using all small RNA sequences, which made it possible to align Fastq files against them and subsequently recover the number of reads aligned to each small RNA using an in-house Python script.

Sequence annotation was performed against the genomes of humans, bacteria, and archaea using Basic Local Alignment Search Tool Nucleotide (BLASTn) [56]. The word size was changed to 15 (-w 15) considering the minimum size of nucleotides in the sequences generated with BrumiR, and only the “best hit” of the results delivered was considered, that is, those hits that presented the lowest e-value.

Downstream and differential expression analyses were performed using the R package, DESeq2. The results with a *p*-value equal to or lower than 0.01 were considered significantly differentially expressed small RNAs [57]. The normalized count was extracted from the DESeq2 analysis to be correlated with LDL-C levels using the Pearson coefficient. Results with a *p*-value equal to or less than 0.05 were considered significant.

Finally, the interaction of differentially expressed small RNAs and the mRNA sequences of twenty molecules considered relevant for cholesterol metabolism in humans (Table 2) was evaluated using IntaRNA v2.3.1 [58], with default parameters. Appendix A summarizes the bioinformatic analysis processes indicated above.

### 4.5. Secondary Structure Prediction

The prediction of the secondary structure of small RNAs was performed using RNAfold WebServer [59,60]. The standard parameters provided by the platform were maintained, except for the option to “avoid isolated base pairs”. Sequences of interest were individually entered to obtain the minimum free energy (MFE) prediction, in addition to the Plain sequence and base pair probability calculation. The above was graphically represented in MFE simple structure drawings, and these were downloaded in .png image format.

## 5. Conclusions

Our results indicate that there were differences in the expression of small RNAs between participants with hypercholesterolemia and normocholesterolemia, and we highlight small RNA 2909606 associated with *C. eutactus* (family *Lachnospiraceae*), which was upregulated in hypercholesterolemic participants. In this same study group, it was also possible to positively correlate small RNA ID 2149569 from the *B. wexlerae* bacterium with elevated LDL-C levels. In addition, it is evident that some small RNAs present a strong complementarity with LDLR mRNA, with this being an important receptor associated with cholesterol metabolism in humans. Although there are other molecules involved in cholesterol metabolism, we hope to obtain more results regarding these in the near future. To deepen our findings, more functional analyses are required to validate the results associated with small RNA 2909606 and other small RNAs indicated in this work. The use of cell cultures of hepatoma cells could make it possible to establish the expression levels of LDLR, as well as other cellular molecules of importance in the metabolism of human cholesterol, revealing the direct impact of the described small RNAs. Likewise, we intend to analyze the non-small RNA readings obtained from the sequencing of stool samples to better characterize the different types of RNAs present at the intestinal level in these participants. Our results provide new evidence regarding the prediction of the interaction of RNAs among different species, specifically associating the impact of gut microbiota with cholesterol metabolism, which we hope can be a contribution to understanding the modulation of the metabolism of this molecule with physiological and pathological importance with the control of gut microorganisms and their small RNAs. This becomes relevant when considering the role of cholesterol in cardiovascular diseases and other metabolic diseases and seeking to contribute to a precision medicine approach to be sought in the future in terms of the prevention and treatment of cardiovascular diseases.

## Figures and Tables

**Figure 1 ijms-24-07213-f001:**
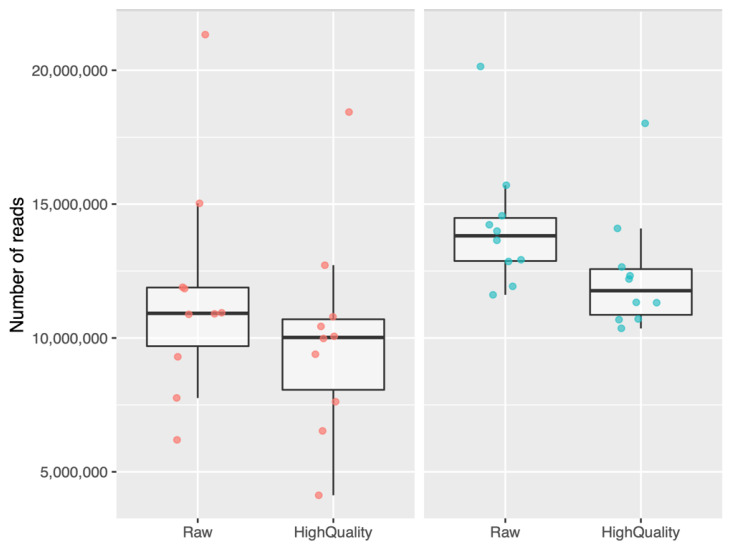
Quantity of sequences obtained in hypercholesterolemic and normocholesterolemic participants. Raw, raw streams in fastp; HighQuality, filtered sequences and high-quality (at least Q30) post-processing with Fastp. Red dots, hypercholesterolemic; blue dots, normocholesterolemic.

**Figure 2 ijms-24-07213-f002:**
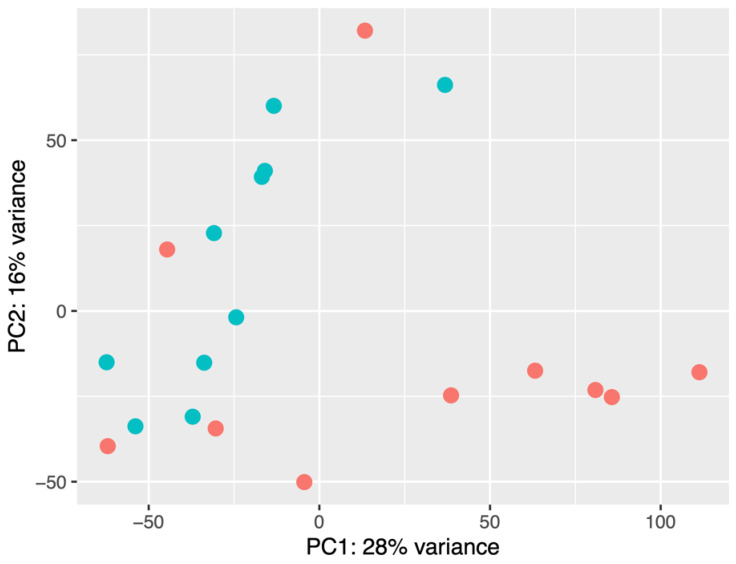
PCA of normocholesterolemic (*n* = 10) and hypercholesterolemic (*n* = 10) participants. Red dots: hypercholesterolemic. Blue dots: normocholesterolemic.

**Figure 3 ijms-24-07213-f003:**
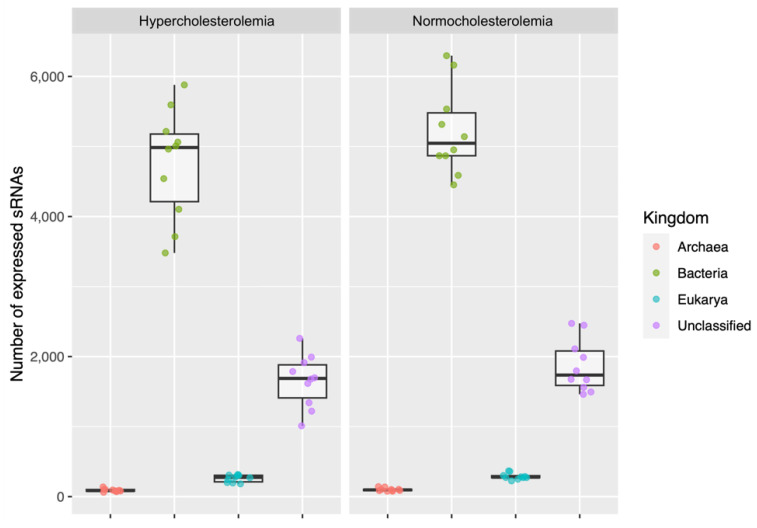
Small RNAs from bacteria, archaea, and eukaryotes, and unclassified ones associated with participants with hypercholesterolemia and normocholesterolemia.

**Figure 4 ijms-24-07213-f004:**
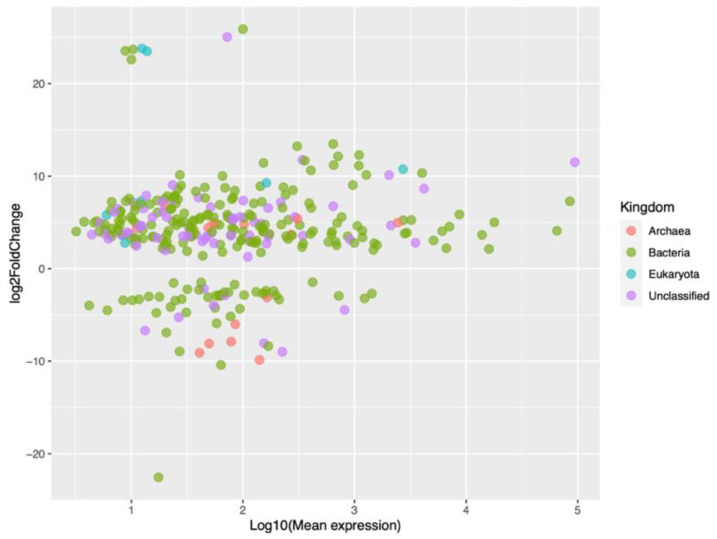
MA plot of small RNAs from bacteria, archaea, and eukaryotes, and unclassified ones.

**Figure 5 ijms-24-07213-f005:**
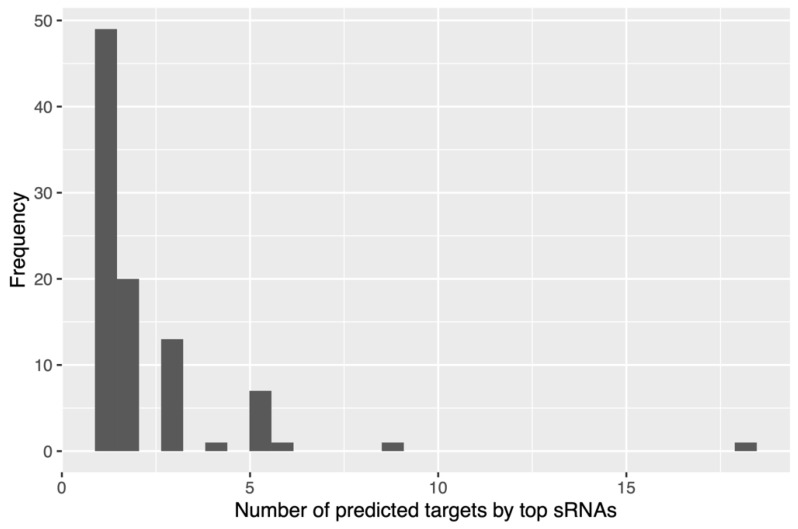
Frequency of small RNAs obtained with respect to mRNAs of genes associated with cholesterol.

**Figure 6 ijms-24-07213-f006:**
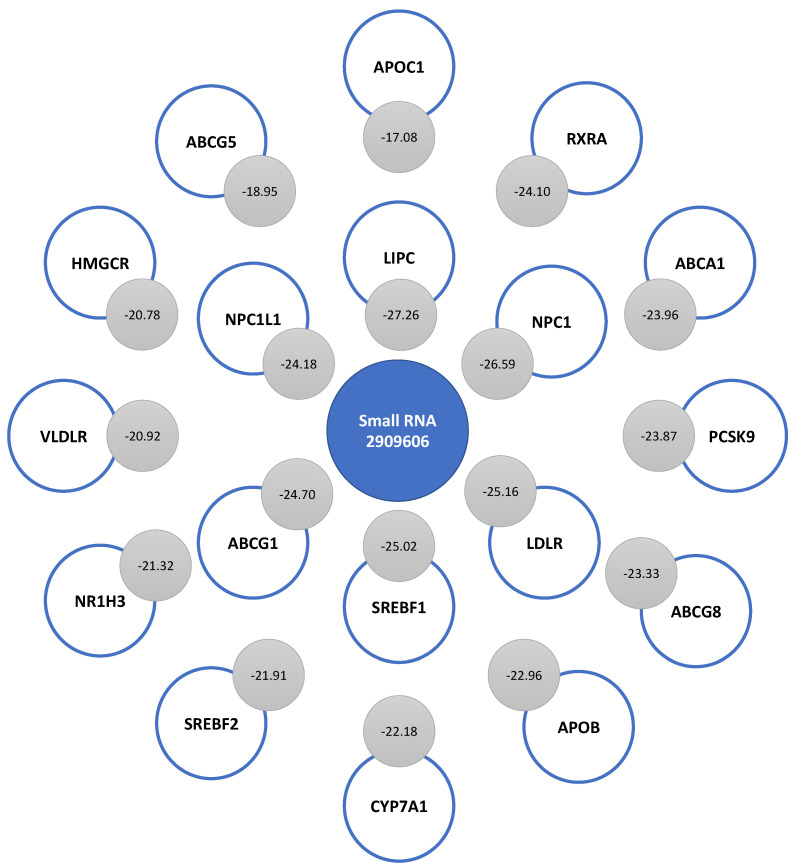
Complementarity of the sequence of small RNA 2909606 with 18 mRNAs of genes involved in cholesterol metabolism. The binding energy levels associated with each interaction are indicated in the gray circles.

**Figure 7 ijms-24-07213-f007:**
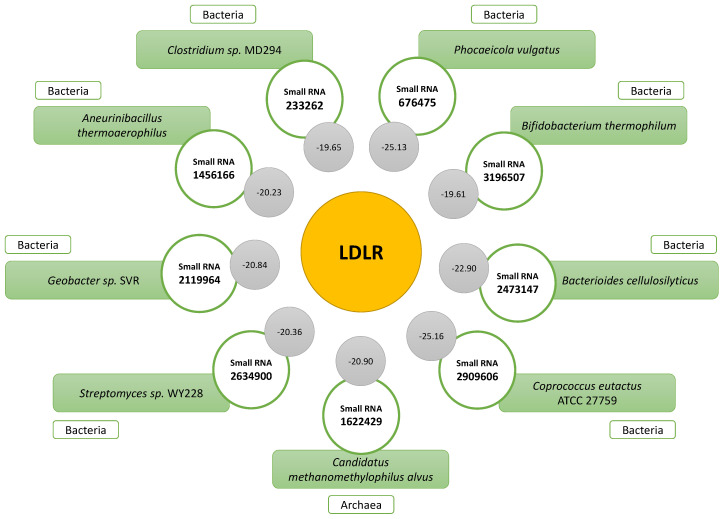
Small RNAs showing significant complementarity with mRNA LDLR. The binding energy levels associated with each interaction are indicated in the gray circles.

**Figure 8 ijms-24-07213-f008:**
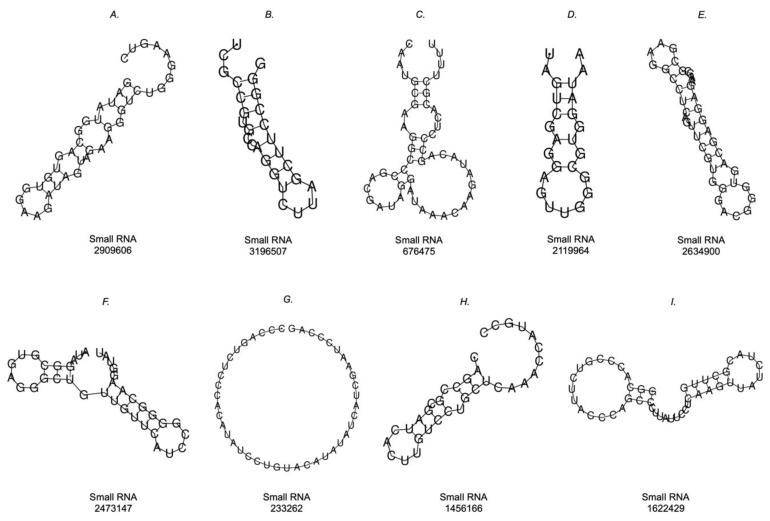
Prediction of the secondary structures of small RNAs that interacted with mRNA LDLR. Simple MFE structure drawings were obtained with RNAfold WebServer.

**Table 1 ijms-24-07213-t001:** Anthropometric and biochemical characteristics of hypercholesterolemic and normocholesterolemic individuals.

Parameter	Normocholesterolemic (*n* = 30)	Hypercholesterolemic (*n* = 27)	*p*
Age (years)	60.07 ± 10.27	60.83 ± 5.00	0.525
Height (m)	1.64 ± 0.02	1.58 ± 0.02	0.068
Weight (kg)	79.34 ± 2.61	64.48 ± 1.83	<0.0001 *
BMI ^1^ (kg/m^2^)	29.72 ± 4.20	25.83 ± 3.21	0.004 *
SBP ^2^ (mmHg)	128.00 ± 16.77	123.00 ± 10.19	0.765
DBP ^3^ (mmHg)	81.29 ± 13.97	80.42 ± 8.93	0.970
Glucose (mmol/L)	5.88 ± 0.79	5.79 ± 0.72	0.688
Triglycerides (mmol/L)	1.73 ± 0.97	2.26 ± 0.93	0.003 *
Total Cholesterol (mmol/L)	4.40 ± 0.53	7.55 ± 0.73	<0.0001 *
HDL ^4^-Cholesterol (mmol/L)	1.51 ± 0.31	1.58 ± 0.40	0.827
LDL ^5^-Cholesterol (mmol/L)	2.24 ± 0.41	5.23 ± 0.74	<0.0001 *

^1^ BMI: body mass index. ^2^ SBP: systolic blood pressure. ^3^ DBP: diastolic blood pressure. ^4^ HDL: high-density lipoprotein. ^5^ LDL: low-density lipoprotein. Mann–Whitney test. * *p* < 0.05.

**Table 2 ijms-24-07213-t002:** Molecules associated with human cholesterol metabolism.

NCBI Code (mRNA)	Molecule Name	Abbreviation
XM_011518339.3	ATP binding cassette subfamily A member 1	ABCA1
XM_011529806.1	ATP binding cassette subfamily G member 1	ABCG1
XM_011533024.2	ATP binding cassette subfamily G member 5	ABCG5
XM_011533029.2	ATP binding cassette subfamily G member 8	ABCG8
NM_000039.3	Apolipoprotein A1	APOA1
NM_000384.3	Apolipoprotein B	APOB
NM_001321065.2	Apolipoprotein C1	APOC1
NM_001302688.2	Apolipoprotein E	APOE
NM_000780.4	Cytochrome P450 family 7 subfamily A member 1	CYP7A1
XM_011543357.1	3-hydroxy-3-methylglutaryl-CoA reductase	HMGCR
NM_000527.5	Low density lipoprotein receptor	LDLR
XM_005254374.4	Lipase C	LIPC
XM_005258277.1	NPC intracellular cholesterol transporter 1	NPC1
XM_011515326.3	NPC1 like intracellular cholesterol transporter 1	NPC1L1
NM_005693.4	Nuclear receptor subfamily 1 group H member 3	NR1H3
NM_174936.4	Proprotein convertase subtilisin/kexin type 9	PCSK9
NM_002957.6	Retinoid X receptor alpha	RXRA
NM_003383.5	Very low-density lipoprotein receptor	VLDLR
XM_024450893.1	Sterol regulatory element binding transcription factor 1	SREBF1
XM_011530347.2	Sterol regulatory element binding transcription factor 2	SREBF2

## Data Availability

The data presented in the study are deposited in the BioProject repository, accession number PRJNA949433.

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
