# Peer review of "Differences in Bacterial Small RNAs in Stool Samples from Hypercholesterolemic and Normocholesterolemic Subjects"

_ijms, 2023, doi:10.3390/ijms24087213_

Round 1

Reviewer 1 Report

In the present work, the authors introduced significant differences in bacterial small RNAs associated with cholesterol metabolism in hypercholesterolemic and normocholesterolemic participants. They used RNA sequencing techniques and different bioinformatic analyses to reach their conclusion. They figured out that small RNA 2909606 is specifically upregulated in hypercholesterolemic participants' stool samples. They also claimed that small RNA 2909606 and other small RNAs can interact with LDL receptors. Generally, the authors presented interesting work and results. However, the results need to be supported by experimental molecular work to confirm their results and conclusion. If they can add more results in the lab. the value of the presented work will be enhanced more.

For example; they can introduce small RNA 2909606 - or another one- into Huh7 hepatoma cells and check the expression levels of the predicted target genes according to their results, at least the expression level of LDLR. They also can estimate LDL levels.

Moreover, the author should explain and discuss how the small bacterial RNA moved from the gut and inter the hepatic cells to affect cholesterol biosynthesis. Such information is important to the readers.

Author Response

  • In the present work, the authors introduced significant differences in bacterial small RNAs associated with cholesterol metabolism in hypercholesterolemic and normocholesterolemic participants. They used RNA sequencing techniques and different bioinformatic analyses to reach their conclusion. They figured out that small RNA 2909606 is specifically upregulated in hypercholesterolemic participants' stool samples. They also claimed that small RNA 2909606 and other small RNAs can interact with LDL receptors. Generally, the authors presented interesting work and results. However, the results need to be supported by experimental molecular work to confirm their results and conclusion. If they can add more results in the lab. the value of the presented work will be enhanced more.

For example; they can introduce small RNA 2909606 - or another one- into Huh7 hepatoma cells and check the expression levels of the predicted target genes according to their results, at least the expression level of LDLR. They also can estimate LDL levels.

We appreciate the reviewer suggestions about the use of cell culture to confirm the findings of our article and we agree with the importance of these assays. However, this would be difficult to complete in a short time because the analyses were performed during the doctoral thesis project of Cristian Morales using financial support that currently is not available. So, we planned to give continuity to this interesting work as soon as possible by looking for new financial support.

  • Moreover, the author should explain and discuss how the small bacterial RNA moved from the gut and inter the hepatic cells to affect cholesterol biosynthesis. Such information is important to the Reader.

According to the reviewer's recommendation, the way in which bacterial small RNAs can be transported to human cells to exert their effect has been discussed (lines 550-559).

Reviewer 2 Report

This study attempts to elucidate the potential role that small RNA may play in hypercholesterolemic and normocholesterolemic individuals from the perspective of differences in small RNA levels in gut microbiota. I think this study is innovative. Therefore, I recommend this study for publication in the journal of International Journal of Molecular Sciences after some major revisions.

 Page 3, please change “low-density lipoprotein (LDL) cholesterol (LDL-C)” into “low-density lipoprotein (LDL) cholesterol (C)”.

Page 3-page 4, the introduction section is not written concisely enough, and it is suggested to remove the content about small RNAs in plants. Focus on the background knowledge of small RNAs in gut microbiota related to host health.

Page 7, Figure 1 does not make much sense and is recommended to be placed in the supplementary material.

Page 8, “From the hypercholesterolemic (n=27) and normocholesterolemic (n=30) groups, the 10 participants with the highest levels of LDL-C and the 10 participants with the lowest levels of this biomolecule were selected.” Please explain why participants with the highest or lowest LDL-C levels were selected over those with intermediate LDL-C levels.

Page 9, It is suggested to add a correlation analysis to account for the relationship between small RNAs and LDL-C.

Page 11 and 12, The names of microorganisms (family name) should be italicized.

Page 16, please delete the references in the last paragraph.

Page 14 and 15, please revise the first two paragraphs of the discussion part to be more concise.

Author Response

  • Page 3, please change “low-density lipoprotein (LDL) cholesterol (LDL-C)” into “low-density lipoprotein (LDL) cholesterol (C)”.

The sentence was changed (lines 68-69).

  • Page 3-page 4, the introduction section is not written concisely enough, and it is suggested to remove the content about small RNAs in plants. Focus on the background knowledge of small RNAs in gut microbiota related to host health.

The introduction was corrected by excluding those related to small RNAs and plants.

  • Page 7, Figure 1 does not make much sense and is recommended to be placed in the supplementary material.

As suggested, the figure 1 has been placed in the supplementary materials section (Supplementary Figure F1).

  • Page 8, “From the hypercholesterolemic (n=27) and normocholesterolemic (n=30) groups, the 10 participants with the highest levels of LDL-C and the 10 participants with the lowest levels of this biomolecule were selected.” Please explain why participants with the highest or lowest LDL-C levels were selected over those with intermediate LDL-C levels.

As suggested, we explained why the participants with the most extreme levels of LDL-C were selected (lines 282-287).

  • Page 9, It is suggested to add a correlation analysis to account for the relationship between small RNAs and LDL-C.

The correlation analysis between the small RNAs and the LDL-C levels was incorporated according to the reviewer's suggestion (Supplementary Table S3).

  • Page 11 and 12, The names of microorganisms (family name) should be italicized.

Families of microorganisms have been placed in italics.

  • Page 16, please delete the references in the last paragraph.

References to the last paragraph of the discussion have been removed.

  • Page 14 and 15, please revise the first two paragraphs of the discussion part to be more concise.

Relevant information from the first two paragraphs of the discussion was selected and merged into a single paragraph (lines 458-466).

Round 2

Reviewer 1 Report

The manuscript can be accepted and published.